# A randomized pilot trial to evaluate the benefit of the concomitant use of atorvastatin and Raltegravir on immunological markers in protease-inhibitor-treated subjects living with HIV

Eugènia Negredo[1,2☯]*, Montse Jiménez[3☯], Jordi Puig[1], Cora Loste[1], Núria Pérez-Álvarez[1,4], Victor Urrea[3], Patricia Echeverría[1], Anna Bonjoch[1], Bonaventura Clotet[1,2,3], Julià Blanco[2,3]

1 Lluita Contra la SIDA Foundation, Germans Trias i Pujol Research Institute-IGTP, Hospital Universitari Germans Trias i Pujol, Universitat Autònoma de Barcelona, Badalona, Catalonia, Spain, 2 Infectious Diseases and Immunity, Centre for Health and Social Care Research (CESS), Faculty of Medicine, University of Vic–Central University of Catalonia (UVic–UCC), Catalonia, Spain, 3 AIDS Research Institute-IrsiCaixa, Germans Trias i Pujol Research Institute-IGTP, Hospital Universitari Germans Trias i Pujol, Barcelona, Catalonia, Spain, 4 Statistics and Operation Research Department, Technical University of Catalonia, Barcelona, Catalonia, Spain

☯ These authors contributed equally to this work.
* enegredo@flsida.org

## Abstract

### Objective

Optimization of antiretroviral therapy and anti-inflammatory treatments, such as statins, are among the strategies aimed at reducing metabolic disorders, inflammation and immune activation in people living with HIV (PLWH). We evaluated the potential benefit of combining both strategies.

### Design

Forty-two PLWH aged ≥40 years receiving a protease inhibitor (PI)-based regimen were randomized (1:1) to switch from PI to Raltegravir (n = 20), or to remain on PI (n = 22). After 24 weeks, all patients received atorvastatin 20mg/day for 48 weeks.

### Methods

We analyzed plasma inflammatory as well as T-cell maturation, activation, exhaustion and senescence markers at baseline, 24 and 72 weeks,

### Results

Plasma inflammatory markers remained unchanged. Furthermore, no major changes on T-cell maturation subsets, immunoactivation, exhaustion or immunosenescence markers in both CD4 and CD8 T cell compartments were observed. Only a modest decrease in the

**Data Availability Statement:** The results are part of a clinical trial approved by the Hospital Germans

Trias I Pujol Ethics Committee. The Hospital Germans Trias I Pujol Ethics Committee has imposed restrictions on sharing the data publicly. For protection of participants, the information was collected under informed consent. If the researchers objectives fit with the consent and/or an amendment can be sent to the Ethics Committee. The data can be made available after contacting with the Data Manager Sílvia Gel; email: sgel@fls.org

**Funding:** This study was funded by Merk Sharp & Dome (MSD) through the MSD Investigator Studies Program (IIS # 52754), grant to EN. Sponsors or funders did not play any role in the study design, data collection and analysis, decision to publish, or preparation of the manuscript.

**Competing interests:** Unrelated to this work, JB is CEo founder and shareholder of AlbaJuna Therapeutics, S.L. This study was funded by Merk Sharp & Dome (MSD) through the MSD Investigator Studies Program (IIS # 52754), grant to EN. This funder did not play any role in the study design, data collection and analysis, decision to publish, or preparation of the manuscript "This does not alter our adherence to PLOS ONE policies on sharing data and materials.

frequency of $CD38^+$ CD8 T cells and an increase in the frequency of $CD28^-CD57^+$ in both CD4 and CD8 T-cell compartments were noticed in the Raltegravir-switched group.

## Conclusions

The study combined antiretroviral switch to Raltegravir and Statin-based anti-inflammatory strategies to reduce inflammation and chronic immune activation in PLWH. Although this combination was safe and well tolerated, it had minimal impact on inflammatory and immunological markers.

## Clinical Trials Registration

NCT02577042.

## Introduction

The incidence of non-AIDS-defining and age-related comorbidities is increasing among people living with HIV (PLWH) and seems to be higher than in the uninfected control population [1–4]. Although the potential pathogenic mechanisms behind this observation are diverse, three main factors might explain the accentuated aging in this population [5]. Firstly, antiretroviral therapy has enlarged the life expectancy of PLWH, increasing in turn the number of elderly individuals in this population. Nowadays, approximately one-half of the PLWH in the United States are aged 50 or older [6]. Aging itself is a condition associated with a chronic inflammation and immune senescence (inflammaging), which are major contributors to increased prevalence of morbidities [7]. Secondly, natural inflammaging is exacerbated in treated PLWH by the persistent inflammatory status and activation of the immune system induced by HIV infection *per se* [8, 9], increasing the risk of age-related morbidities [10–13]. Finally, the continuous exposure to antiretrovirals and their toxicities, mainly but not exclusively associated to former antiretroviral drugs [14], may contribute to accelerate the emergence of some of age-related diseases such as dyslipidemia, cardiovascular events, renal damage or low bone mineral density [15].

Consequently, one of the current aims of the clinical management of PLWH is the control of chronic non-AIDS-related comorbidities in an increasingly older and complex population [5]. Among clinical challenges, the maintenance of viral suppression is a premise, as HIV undetectability is a key factor to reduce the systemic inflammation [16], and can be now safely achieved by using newest and less toxic antiretroviral drugs, particularly in the elderly population. However, virological suppression is not sufficient to eliminate chronic inflammation, as appropriately treated PLWH still show higher levels of inflammation and immune activation markers in comparison with general population [8]. Among these markers, IL-6 seems to be a major player of inflammaging [17] and a determinant factor of survival in PLWH [12].

Additional strategies have been evaluated to curtail the chronic inflammatory status. These strategies encompass therapy intensification to reduce residual viral replication [18], switch antiretrovirals to safer profiles to reduce related toxicities [19], and addition of immunomodulatory therapies to specifically reduce inflammation or immune activation, including anti-CMV drugs [20], immunosuppressors [21] or statins [22]. Despite these wide clinical efforts, no conclusive data have been generated, and only some switch strategies but not immune-focused approaches have impacted current clinical guidelines.

A beneficial effect has been demonstrated with raltegravir, an integrase inhibitor. It exhibits a rapid, potent and durable antiretroviral activity with a better impact on bone mineral density and lipid and renal profiles than from a protease inhibitors (PIs) [23–26]. In addition, integrase inhibitors might have a benefit on immune and inflammatory parameters in comparison with other antiretroviral classes [27–29]. Similarly, statins are lipid-lowering drugs that also exert anti-inflammatory effects, and have immune-modulatory properties. Recent studies in HIV-infected population have suggested that statins impact inflammatory markers [22, 30–35], and that its use is associated with a lower risk of non-AIDS defining morbidities and malignancies and mortality [36–39]. Nonetheless, data are limited and mainly emerged from retrospective or cohort studies. Currently, a prospective, randomized, placebo-controlled trial (REPRIEVE study) is ongoing to assess a statin strategy for the primary prevention of major cardiovascular events [40]. Although around 7,500 patients have already been included from over 120 clinical sites across 12 countries, final results will be available in several years.

Meanwhile, based on these data, we explored the potential benefits of combining both approaches in the Ralator study, which includes a double strategy to reduce systemic inflammation and improve lipid profile in chronic HIV population by i) switching the PI to Raltegravir; and ii) adding atorvastatin for one year, owed their potential anti-inflammatory effect.

## Methods

### Study design and population

Ralator is a 72-week, active-controlled, single site, randomized, open-label, pilot study in PLWH aged 40 years or over (registered under the code NCT02577042 at the Clinical trials website). Candidates to be included in the study were PLWH, aged $\geq$ 40 years, receiving a PI-based antiretroviral regimen combined with tenofovir/emtricitabine or abacavir/lamivudine for at least 6 months and maintaining undetectable plasma HIV-1 RNA (VL < 50 copies/mL) for at least 12 months. Exclusion criteria were history of virological failure to integrase inhibitors and suspected or documented resistance mutations to the integrase or NRTI, systemic concurrent processes (e.g. active coinfection with hepatitis C or B, acute systemic infection within the last 4 months, neoplasia or chronic inflammatory process), treatment with anti-inflammatory, anticoagulant or antiplatelet drugs (e.g. corticosteroids, aspirin or other) and therapy with statins and raltegravir within the last 6 months. The ethics committee of the Hospital Germans Trias i Pujol and the Spanish Medicines Agency (AEMPS) approved the protocol, and the study was performed in accordance with the principles of the Declaration of Helsinki. All participants provided written informed consent before inclusion.

At randomization, patients were stratified according to: 1) the baseline levels of LDL-cholesterol, cutoff value 160 mg/dL since a LDL-cholesterol 130–159 mg/dL is considered borderline high for people without cardiovascular risk factors and 2) the nucleoside drugs used, tenofovir/emtricitabine or abacavir/lamivudine. Patients were randomly assigned in a 1:1 ratio to: 1) continue with the same PI-based regimen, including tenofovir/emtricitabine or abacavir/lamivudine, for 24 weeks (Control group); or 2) switch the PI to Raltegravir (400mg/12 hours), plus tenofovir/emtricitabine or abacavir/lamivudine, for 24 weeks (Raltegravir group). The assignation was centralized and performed by telephone. After 24 weeks, atorvastatin (20mg/day), was added to both Control and Raltegravir groups for 48 weeks (Fig 1). All data was collected at the Hospital Germans Trias i Pujol.

### Study objective and endpoints

The primary objective of the study was to compare changes in plasma levels of IL-6 between Control and Raltegravir groups. To assess the effect of switching to raltegravir, differences between

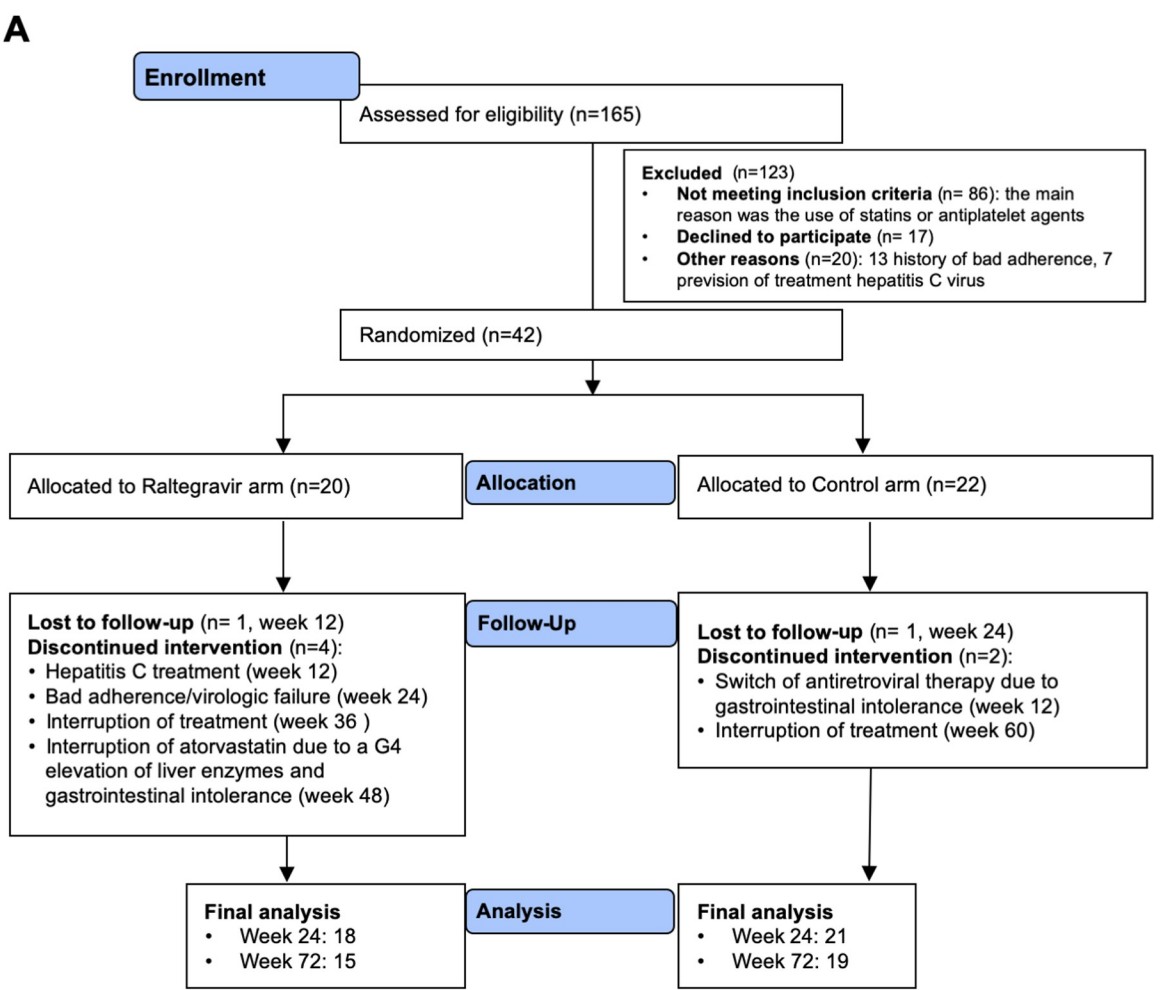

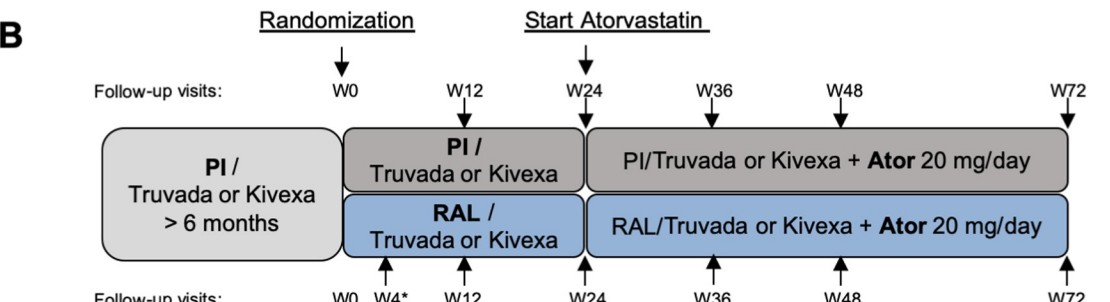

**Fig 1. Trial design. A.** Consolidated Standards of Reporting Trials (CONSORT) flow diagram for the trial showing Participant flow in Control and Raltegravir groups. **B.** Schematic representation of intervention, patients were randomized to the Control and Raltegravir groups and, after 24 weeks, atorvastatin was added to all participants. Follow up visits are indicated for each group.

groups at week 24 and intragroup longitudinal changes at week 24 from baseline were assessed. In addition, we compared both groups at week 72 to assess the effect of adding atorvastatin. Furthermore, we analyzed intragroup longitudinal changes at week 72 from baseline and from week 24.

Secondary objectives were to compare changes in the above-mentioned time periods in: i) other plasma inflammatory/coagulation markers: soluble CD14 (sCD14), C-reactive protein (CRP) and D-Dimer; ii) CD4 and CD8 T-cell maturation/activation/exhaustion and immuno-senescence markers; and iii) plasma lipid and lipoprotein parameters (total, HDL-, LDL-cholesterol and triglyceride levels). In addition, participants were followed-up for virological and immunological response by assessing plasma viral load and circulating CD4 and CD8 T cell counts at the indicated timepoints (Fig 1).

Demographic and HIV infection-related data were collected in order to characterize the study population (gender, age, time since HIV diagnosis, time since ART treatment, route of transmission of HIV).

## Plasma inflammation markers

Plasma levels of several markers were measured using ELISA kits following the manufacturer's instructions: C-reactive protein (CRP) and D-dimer from Ray Biotech; soluble CD14 (sCD14) from Diaclone and IL-6 from ThermoFisher.

## Immunophenotyping and activation markers in peripheral CD4+ and CD8 + T cells

Cryopreserved PBMC ($10x10^6$ cells) were thawed, sequentially washed in RPMI medium (Invitrogen) containing 60%, 20% and 10% of Fetal Bovine Serum (FBS, Invitrogen) and incubated for 15 minutes at 37˚C in RPMI medium supplemented with 10% FBS. PBMC were incubated with Fixable Viability Stain 780 (BD Bioscience, final dilution 1:4000) as described [19]. Then, cells were stained with the following antibodies: CD3 BV605 (clone SK7), CD4 FITC (clone RPA-T4), CD8 BV510 (clone SK1), CD45RA Alexa Fluor 700 (clone HI100), CD197 PE-CF594 (clone 150503), CD57 APC (clone NK-1), HLA-DR BV650 (clone G46-6), CD279(PD-1) BV421 (clone EH12.1), CD38 PE (clone HIT2), CD28 PerCP Cy-5.5 (clone L293) and CD27 BV786 (clone L128) all from BD Bioscience (San Jose, CA). After 15 minutes incubation in the dark at room temperature, cells were washed twice in PBS containing 1% FCS and fixed in formaldehyde 1% in PBS. All samples were acquired in a BD LSRFortessa flow cytometer (BD Bioscience) and further analyzed with FlowJo software (Tree Star Software). Briefly, a time gate was defined to ensure homogenous cell acquisition, then singlets and lymphocyte gates were defined by morphological parameters. Viable CD3+ cells were gated for CD4 and CD8 compartment analyses gated as CD4+CD8- or CD8+CD4-, while double-positive cells and double-negative cells were excluded from the analysis. Further automated detection of marker cutoffs based in fluorescence controls was performed using OurFlow, a custom pipeline running concatenated R packages [19]. Gating strategy is described in S1 Fig. T-cell maturation was analyzed using CD45RA and CD197 (CCR7) expression to define naïve ($T_N$, CD45RA+CCR7−), central memory ($T_{CM}$, CD45RA−CCR7+), Effector ($T_{EF}$, CD45RA−CCR7−) and effector memory RA+ cells ($T_{EMRA}$, CD45RA+CCR7−); CD27 was used to differentiate $T_{EF}$ cells in transitional memory ($T_{TM}$, CD27+CD4+ or CD27−CD8+) and effector memory ($T_{EM}$, CD27−CD4+ or CD27+CD8+). CD4 and CD8 T cells were also analyzed for the expression of HLA-DR and CD38 to define activated cells (% of HLA-DR+CD38+cells), the expression of PD-1 to measure exhaustion (% of PD-1+ cells) and the expression of CD57 and CD28 to measure immunosenescence (% of CD28-CD57+ cells).

## Statistical analyses

Sample size determination was guided by the number of candidates who could be enrolled in the study according to our patient's data base attended in our HIV Unit, whereby 60 patients

were planned to be included into the study. Since this is a pilot study, power analysis was not conducted in order to calculate sample size.

A general descriptive analysis of all the study variables was done by using mean and standard deviation for normally distributed variables or median and interquartile range for the non-parametric variables; relative frequencies was used for categorical variables.

Differences between groups and longitudinal changes where represented and evaluated using parametric (t-test and paired t-test) and non-parametric tests (Mann-Whitney and Wilcoxon signed-rank tests), according to the variable's distribution, and linear mixed-effects models that were fitted for the most relevant markers. For categorical variables, independence between groups was studied using the Chi-Square or Fisher exact test and time effect was assessed by means of a McNemar tests. The Spearman correlation test was used to evaluate the association between quantitative variables. Longitudinal comparisons were performed by linear mixed effects models.

## Results

### Patient characteristics and clinical follow-up

The recruitment period of the study was 32 weeks. The start date of recruitment was February 2015. The follow-up period started on November 2015 and the end of study date was June 2018. A total of 42 participants were included; of these, 20 were randomized to switch to Raltegravir (Raltegravir group) and 22 to continue the same PI/r-based cART regimen (Control group) (Fig 1). The main characteristics at baseline of both groups are summarized in Table 1. No significant differences among groups were found. The overall mean (standard deviation, SD) age of participants was 50.6 (6.2) years and the mean time with HIV infection was 18.7 (7.2) years. The overall mean (SD) CD4 T cell count was 610 (223) cells/μL and the mean nadir CD4 count was 202 (128) cells/μL (Table 1).

Eight participants discontinued the study protocol early, 5 were from the Raltegravir group: study interruption for personal reasons (week 12), Hepatitis C treatment (week 12), virologic failure secondary to a poor adherence (week 24), interruption of treatment due to patient's decision (week 36) and interruption of atorvastatin due to a grade 4 elevation of liver enzymes and gastrointestinal intolerance (week 48); and 3 from the Control group: interruption of treatment due to PI-related gastrointestinal discomfort (week 12), study interruption for personal reasons (week 24) and interruption of treatment due to patient's decision (week 60). The remaining randomized individuals completed the study protocol (Fig 1).

Three participants ─ all in Raltegravir arm ─ showed a grade 3–4 adverse events possibly related to the study medication: 2 patients showed a grade 3 and grade 4 increases of creatine kinase, which improved without interrupting therapy, and, as mentioned above, one participant showed a grade 4 increase of liver enzymes that resolved after interrupting the atorvastatin.

### Changes in soluble plasma markers of inflammation

Four different inflammation plasma markers (IL-6, sCD14, CRP and D-dimer) were measured at baseline, week 24 and week 72 using commercial ELISA kits. Results, summarized in Table 2, clearly showed that either PI switch to Raltegravir or atorvastatin treatment failed to modify IL-6 plasma levels (primary study objective). Indeed, no differences between groups were observed at week 24 (to evaluate the impact of therapy switch) or at week 72 (to evaluate the impact of statin treatment). Furthermore, no longitudinal intragroup differences in IL-6 plasma concentrations were noticed in the Control or Raltegravir groups (Table 2). Similarly, the analysis of D-dimer, sCD14 and CRP plasma levels, showed minimal and no differences between groups at

**Table 1. Baseline characteristics of participants.**

|  | Control group *(n = 22) | Raltegravir group *(n = 20) |
|---|---|---|
| **Age** (Years) | 51.8 (8.2) | 49.5 (3.5) |
| **Gender** (% Male) | 77% | 85% |
| **Route of HIV infection:** (%) |  |  |
| MSM | 50% | 35% |
| HSM | 18% | 40% |
| Injecting drug use | 23% | 15% |
| Other | 9% | 10% |
| **Time from HIV diagnosis** (Years) | 19.2 (7.9) | 18.2 (6.6) |
| **Current protease inhibitor:** (%) |  |  |
| Atazanavir or atazanavir/ritonavir | 27% | 20% |
| Lopinavir/ritonavir | 18% | 15% |
| Darunavir/ritonavir oe cobicistat | 55% | 60% |
| Other | 0% | 5% |
| **Hepatitis C** (%) | 18% | 10% |
| **CD4 count** |  |  |
| Current (cells/µL) | 640 (253) | 579 (184) |
| Current (% of lymphocytes) | 29 (8) | 29 (8) |
| Nadir CD4 count (cells/µL) | 191 (143) | 211 (116) |
| **CD8 count** |  |  |
| Current (cells/µL) | 1003 (409) | 898 (349) |
| Current (% of lymphocytes) | 46 (10) | 43 (9) |
| **LDL-cholesterol > 160 mg/dL (%)** | 9% | 5% |
| **Lipid profile** (mg/dL) |  |  |
| Total cholesterol | 181 (31) | 189 (35) |
| LDL cholesterol | 104 (27) | 108 (31) |
| Triglycerides | 177 (133) | 159 (106) |

*All parameters are expressed as mean and standard deviation (SD) except for discrete variables that are expressed as percentage (%). No significant differences were found among groups.

LDL, Low-density lipoprotein.

24 and 72 weeks and a lack of longitudinal effect intragroup (Table 2). Thus, therapy switch, atorvastatin treatment or the combination of both strategies failed to modify plasma levels of IL-6 and other soluble markers of inflammation and coagulation in our study.

**Table 2. Changes in plasma soluble markers.**

|  | Control group | | | Raltegravir group | | |
|---|---|---|---|---|---|---|
|  | **Baseline** | **W24** | **W72** | **Baseline** | **W24** | **W72** |
|  | **n = 22** | **n = 21** | **n = 19** | **n = 20** | **n = 18** | **n = 15** |
| **IL-6 (pg/mL)** | 40.0 | 41.1 | 41.7 | 42.5 | 39.3 | 43.2 |
| Median (IQR) | (36.9–53.7) | (36.1–53.2) | (34.7–51.5) | (34.5–50.1) | (34.6–49.8) | (36.8–52.9) |
| **D-Dimer (ng/mL)** | 1990 | 1917 | 1868 | 1743 | 1844 | 2051 |
| Median (IQR) | (1574–2523) | (1552–2244) | (1438–2175) | (1459–1894) | (1547–2340) | (1656–2414) |
| **sCD14 (ng/mL)** | 7588 | 7736 | 8342 | 7579 | 7368 | 7658 |
| Median (IQR) | (7121–8419) | (7307–8465) | (7333–8 856) | (7039–8726) | (7131–9056) | (7039–8355) |
| **CRP (ng/mL)** | 8587 | 7531 | 10268 | 6744 | 7059 | 8334 |
| Median (IQR) | (6700–16735) | (5056–15069) | (5695–21228) | (4461–15211) | (4729–10868) | (5134–17392) |

## Changes in T-cell maturation, activation, exhaustion and immunosenescence in PBMCs

No changes in absolute counts or percentages of CD4 and CD8 T cells, either intragroup or intergroup were observed at any time point. Immunophenotyping of T cells was performed at baseline, week 24 and week 72. First, we explored changes in T cell maturation, by defining five T-cell maturation subsets: naïve ($T_N$), central memory ($T_{CM}$), transitional memory ($T_{TM}$), effector memory ($T_{EM}$) and effector memory RA+ ($T_{EMRA}$) based on the combination of CD197, CD45RA and CD27 markers (see Methods). In CD4+ T cells, we observed that the Control group tended to show higher levels of $T_N$ cells and lower levels of $T_{TM}$ cells at all time-points, including baseline, in comparison with Raltegravir group; (Fig 2A). Longitudinally,

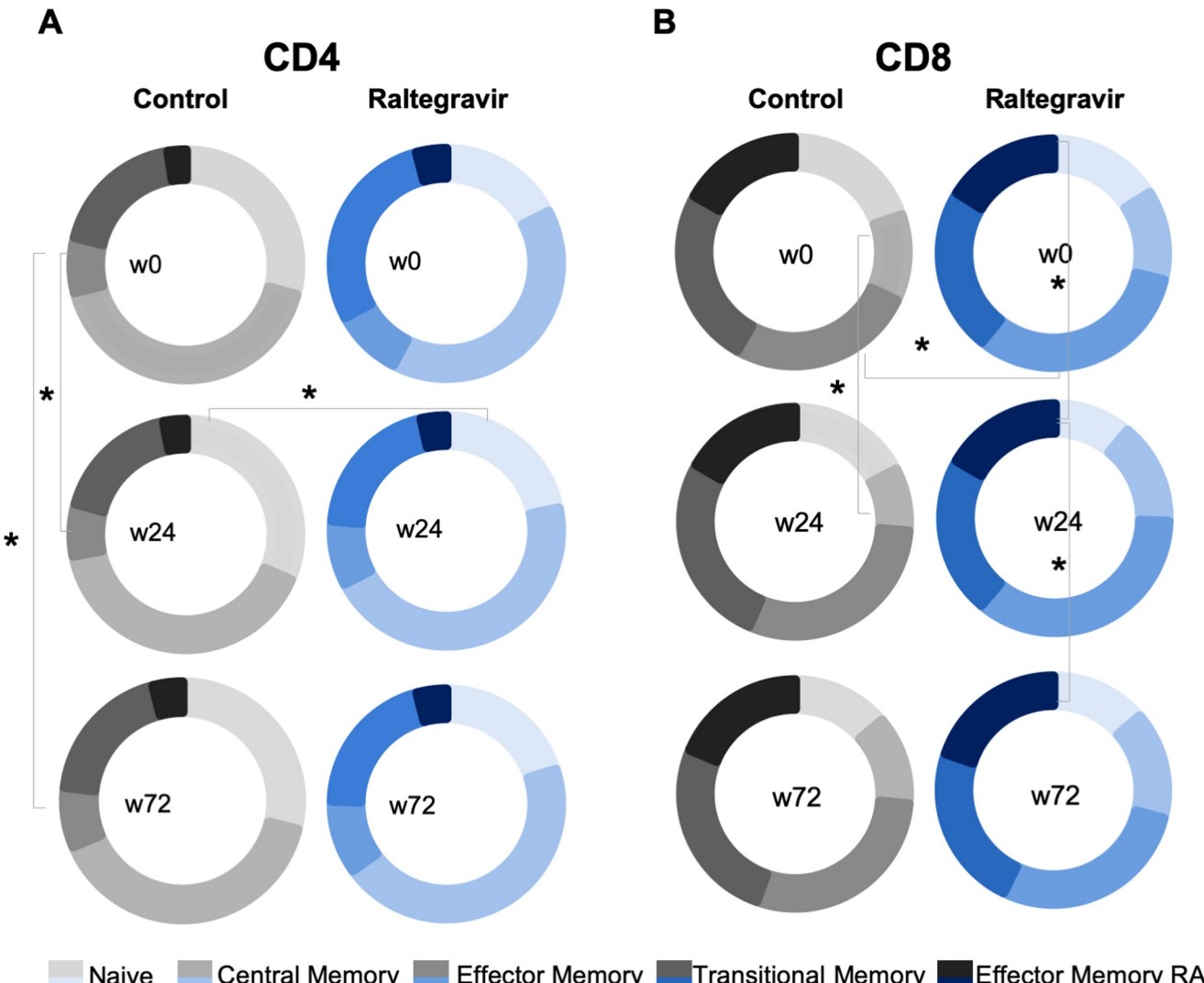

**Fig 2. of maturation subsets in CD4 and CD8 T cells. Analysis** Figure shows median values for the frequency of the maturation subsets (Naïve, Central Memory, Effector Memory, Transitional Memory and Effector Memory RA+) in CD4 (panel A) or CD8 T cells (Panel B) in Control (greyscale) and Raltegravir (blue) groups at three different time points: week 0 (start of the study), week 24 (treatment follow-up), and week 72 (48 weeks after atorvastatin treatment). Maturation stages were defined based on the combination of CD45RA, CD197 (CCR7) and CD27 as described in Methods. Asterisks denote p<0.05 either intergroups or longitudinally intragroup as indicated.

minimal changes were observed, although a decrease in $T_{CM}$ at week 24 and 72 from baseline in Control group was noticed (Fig 2A).

The analysis of maturation subsets in CD8 T cells also showed minimal differences between groups at baseline, although a lower frequency of $T_{TM}$ cells was noticed in the control group. Longitudinal analysis showed a decrease in $T_{CM}$ frequency between baseline and week 24 in the Control group. In addition, the Raltegravir group showed a transient decrease in the frequency of $T_{EMRA}$ CD8 T cells at week 24, that vanished at week 72 in (Fig 2B).

Next, we analyzed changes in immune activation (defined by CD38 expression or CD38 and HLA-DR co-expression), immune exhaustion (defined by PD-1 expression) and immunosenescence (defined by the frequency of CD28-CD57+ cells). In CD4 T cells, consistent with the lower frequency of $T_N$ cells, CD38 expression (which is constitutive in $T_N$ cells) tended to be lower in the Raltegravir group. Longitudinally, no changes were observed in activation (CD38+HLA-DR+) and exhaustion (PD-1+) markers in any group, although an unexpected increase in immunosenescence (CD28-CD57+) was specifically observed in the Raltegravir group at week 72 (Fig 3).

Similarly, minimal changes were noticed in activated (CD38+HLA-DR+) and exhausted (PD-1+) CD8 cells, which did not show any difference inter or intragroup throughout the study. However, CD38 expression decreased between baseline and week 72 in Raltegravir group (p = 0.06, Fig 3), In contrast, we found an increase in the frequency of immunosenescent (CD28-CD57+) CD8 T cells between week 24 and week 72 in the Raltegravir group that was not observed in the Control group (Fig 3).

To ascertain the impact of Raltegravir switch (change at week 24 from baseline) and atorvastatin treatment (change at week 72 from week 24) on the different parameters we have properly transformed data to normalize and analyze them using linear mixed-effects models. These models were fitted in different ways, parameterizing the time variable as continuous and as categorical time points and including or excluding interaction effects. The initial analysis failed to confirm a significant change overtime in the frequency CD28-CD57+ CD4 or CD8 T cells, but confirmed the decay of CD38 expression in CD8 T cells.

## Changes in lipid profile

A decrease in triglyceride levels was seen at week 24 from baseline in the Raltegravir group. At week 72 from baseline (effect of switch antiretroviral and adding atorvastatin), a decrease was seen in both groups in all lipid parameters, as well as at week 72 from week 24 (effect of atorvastatin) (S1 Table). No differences were seen between groups at week 72 in any lipid parameter.

## Discussion

We have evaluated the anti-inflammatory effect of atorvastatin in a prospective clinical trial measuring changes in inflammatory, immune activation and immunosenescence markers after adding atorvastatin, at intermediate dose (20 mg per day), to a Raltegravir- or a PI-based regimen. The selection of atorvastatin was guided by the low risk of drug-drug interactions between this statin and boosted-PI, as well as its low cost. The intermediate dose (20 mg per day) was selected to avoid toxicity.

Four soluble plasma markers that have been shown to predict mortality in treated HIV infection in different clinical settings [41–43] were selected to determine changes on overall inflammation: IL-6, CRP, D-dimer and sCD14. These markers cover pyrogenic inflammation, acute-phase proteins, coagulation and monocyte activation, respectively. Moreover, those markers have been shown to be impacted by at least one of these strategies (statin addition or Raltegravir intensification or switch): Raltegravir has been reported to reduce D-dimer plasma

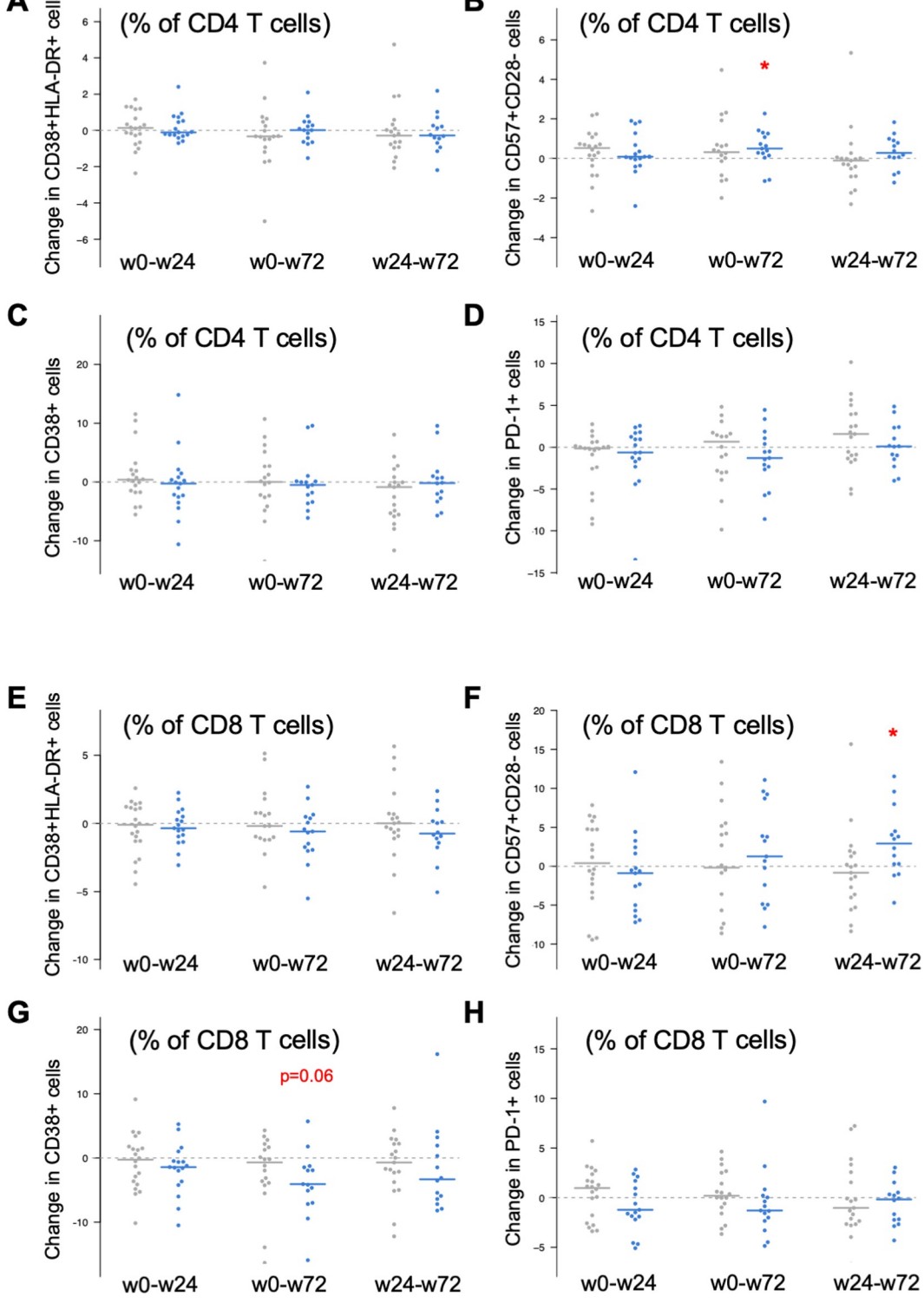

**Fig 3. Evaluation of the changes in activation, CD38+ expression, immunosenescence and exhaustion levels in CD4 and CD8 T cells.** Measure of changes in mean activation levels (defined by HLA-DR+CD38+ coexpression, panels A and E) CD38+ cells (panels B and F) senescence (CD57+CD28−, panels C and G) and exhaustion (PD-1+, panels D and H) levels in CD4+ T cells (panels A-D) and CD8 T cells (panels E-H) comparing both study groups. These differences were evaluated between w0–w24, w0–w72 and w24–w72. Control group is represented in grey and the Raltegravir group in blue. Asterisks denote p<0.05.

levels [44], specifically in patients on a PI-based regimen [8], while statins (pravastatin or pita-vastatin) reduced sCD14 levels [24] or CRP levels [31] in antiretroviral-treated individuals. Despite those reported effects, all selected biomarkers remained unchanged through our study in both arms.

To fully analyze the immunological effects of antiretroviral switching and atorvastatin treatment, we have also evaluated the impact on immunological markers of immune activation, exhaustion and immunosenescence. Consistent with the minimal impact on inflammation, most cellular immunological markers remained unchanged throughout the study. Actually, only minimal changes in the frequency of maturation CD4 and CD8 T cell subsets were noticed, that were consistent with previous reported effects of integrase inhibitors switch [19] but unrelated to the reported effects of Raltegravir intensification [27, 28]. In contrast, among our patients, the PI switch to Raltegravir and the addition of atorvastatin had minimal effect on reducing CD38 expression on CD8 T cells and on increasing the frequency of immunose-nescent CD4 and CD8 T cells. Regarding CD38 expression on CD8 T cells, this effect seems to be consistent with previously reported beneficial effects of Raltegravir intensification [27, 45]. However, this effect has not been observed in PI to integrase inhibitor switch strategies [19], suggesting a direct effect of atorvastatin, that was not fully supported by statistical modelling of changes in CD38 expression. Statins have shown the ability to reduce CD8 activation in PLWH either untreated [22] or under antiretroviral drugs [33]. However, this effect is still controversial, as different studies failed to observe relevant effects of statins in immune activation [46]. Our data are on agreement with these latter reports, although the dose and the specific statin used in the different studies are confounding factors that may impact on the final effect, thus contributing to the conflicting data reported.

Regarding immunosenescence measured by the frequency of CD28-CD57+ cells, we observed an unexpected increase at week 72 compared to week 24 in the CD8 T cell compartment and a similar increase at week 72 compared to baseline in CD4 T cells, both in the Raltegravir group, suggesting an effect of the concomitant use of raltegravir and atorvastatin. Although the changes were low in magnitude, they were consistent in both T cell subsets. This marker is increased in HIV individuals and it is associated with several comorbidities [47]. However, its modulation by antiretroviral therapy has not been fully characterized, and no specific effects of Raltegravir have been identified. Indeed, impact of HIV infection on this marker is qualitatively different from other chronic infection such as cytomegalovirus [48].

Our study has some limitations, including the low sample size and the loss of follow-up of a considerable number of patients. Some study criteria (aged $\geq$ 40 years, receiving a PI) could explain the high difficulty of achieving and maintaining patients in the study since the high genetic barrier makes this antiretroviral class the best option at that time for subjects with not optimal treatment adherence. As well, the low statin dose used to avoid toxicity may also contribute to the lack of impact of the intervention. In addition, the lack of a study arm without statin treatment impedes the direct identification of separate effects of Raltegravir switch and statin treatment. We have overcome this limitation by using longitudinal linear models to evaluate the ultimate responsible for the changes observed, suggesting a direct role of atorvastatin. However, those changes are small and their clinical impact on the long-term uncertain.

Therefore, considering our data, switching from a PI to Raltegravir together with low dose atorvastatin did not provide significant immunological benefits in terms of a reduction in inflammatory or immune activation markers. However, the design of the REPRIEVE study including a large number of patients with a long follow-up will help us to elucidate this issue.

## Supporting information

**S1 Fig.**
(DOCX)

**S1 Table.**
(DOCX)

**S1 Checklist.**
(DOC)

## Author Contributions

**Conceptualization:** Eugènia Negredo, Bonaventura Clotet, Julià Blanco.

**Data curation:** Montse Jiménez, Jordi Puig, Núria Pérez-Álvarez, Patricia Echeverría, Anna Bonjoch.

**Formal analysis:** Montse Jiménez, Jordi Puig, Núria Pérez-Álvarez, Victor Urrea.

**Funding acquisition:** Eugènia Negredo.

**Investigation:** Jordi Puig, Cora Loste, Patricia Echeverría, Anna Bonjoch, Bonaventura Clotet, Julià Blanco.

**Methodology:** Julià Blanco.

**Project administration:** Cora Loste.

**Supervision:** Eugènia Negredo.

**Validation:** Eugènia Negredo.

**Writing – original draft:** Julià Blanco.

**Writing – review & editing:** Eugènia Negredo, Montse Jiménez, Jordi Puig, Cora Loste, Núria Pérez-Álvarez, Victor Urrea, Patricia Echeverría, Anna Bonjoch, Bonaventura Clotet, Julià Blanco.

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
