## [Decision Letter · Decision Letter 0]

13 Mar 2020

PONE-D-20-01200

Limited Benefit of the Concomitant Use of Atorvastatin and Raltegravir on Immunological Markers in Treated Subjects living with HIV.

PLOS ONE

Dear Dr Negredo,

Thank you for submitting your manuscript to PLOS ONE. After careful consideration, we feel that it has merit but does not fully meet PLOS ONE’s publication criteria as it currently stands. Therefore, we invite you to submit a revised version of the manuscript that addresses the points raised during the review process.

We would appreciate receiving your revised manuscript by Apr 27 2020 11:59PM. To enhance the reproducibility of your results, we recommend that if applicable you deposit your laboratory protocols in protocols.io, where a protocol can be assigned its own identifier (DOI) such that it can be cited independently in the future. For instructions see: http://journals.plos.org/plosone/s/submission-guidelines#loc-laboratory-protocols

We look forward to receiving your revised manuscript.

Kind regards,

Alan Winston

Academic Editor

PLOS ONE

Journal Requirements:

2) PLOS requires an ORCID iD for the corresponding author in Editorial Manager on papers submitted after December 6th, 2016. Please ensure that you have an ORCID iD and that it is validated in Editorial Manager. To do this, go to ‘Update my Information’ (in the upper left-hand corner of the main menu), and click on the Fetch/Validate link next to the ORCID field. This will take you to the ORCID site and allow you to create a new iD or authenticate a pre-existing iD in Editorial Manager. Please see the following video for instructions on linking an ORCID iD to your Editorial Manager account: https://www.youtube.com/watch?v=_xcclfuvtxQ

3) Please note that all PLOS journals ask authors to adhere to our policies for sharing of data and materials: https://journals.plos.org/plosone/s/data-availability. According to PLOS ONE’s Data Availability policy, we require that the minimal dataset underlying results reported in the submission must be made immediately and freely available at the time of publication. As such, please remove any instances of 'unpublished data' or 'data not shown' in your manuscript and replace these with either the relevant data (in the form of additional figures, tables or descriptive text, as appropriate), a citation to where the data can be found, or remove altogether any statements supported by data not presented in the manuscript.

4) Please include captions for your Supporting Information files at the end of your manuscript, and update any in-text citations to match accordingly. Please see our Supporting Information guidelines for more information: http://journals.plos.org/plosone/s/supporting-information.

5)  Thank you for stating the following in the Financial Disclosure section:

[This study was funded by Merk Sharp & Dome (MSD) through the MSD Investigator

Studies Program (IIS # 52754), grant to EN.

Sponsors or funders did not play any role in the study design, data collection and analysis, decision to publish, or preparation of the manuscript.].

We note that you received funding from a commercial source: Merk Sharp & Dome

6) Please amend either the title on the online submission form (via Edit Submission) or the title in the manuscript so that they are identical.

7) We note that you have indicated that data from this study are available upon request. PLOS only allows data to be available upon request if there are legal or ethical restrictions on sharing data publicly. For information on unacceptable data access restrictions, please see http://journals.plos.org/plosone/s/data-availability#loc-unacceptable-data-access-restrictions.

Reviewers' comments:

Reviewer's Responses to Questions

**Comments to the Author**

1. Is the manuscript technically sound, and do the data support the conclusions?

Reviewer #1: Yes

Reviewer #2: Partly

2. Has the statistical analysis been performed appropriately and rigorously? 

Reviewer #1: Yes

Reviewer #2: No

3. Have the authors made all data underlying the findings in their manuscript fully available?

Reviewer #1: Yes

Reviewer #2: No

4. Is the manuscript presented in an intelligible fashion and written in standard English?

Reviewer #1: Yes

Reviewer #2: Yes

5. Review Comments to the Author

Reviewer #1: Negredo et al present a pilot trial to assess the concomitant use of atorvastatin and raltegravir on immunological markers in treated subjects living with HIV switching from PI.

The idea to combine two potentially effective strategies options is original and merit consideration for publication, regardless negative results in reducing inflammation and immune activation.

Some issues should be considered to improve understanding of methods and overall discussion.

1. Title should mention that PLWH were PI-experienced.

2. Inclusion criteria should specify if all the patients were Raltegravir and statine naïve.

3. Table 1 should mention the cardiovascular risk profile of the patients and the proportion of the individuals offered statine regardless of dyslipidemia or low CVR.

4. It should be mention why a LDL cut off of 160 mg was chosen for stratification and be consistent through the paper when using mg of mmol units .

5. Line 284-294 is in fact part of the background and can be summarized in this section.

6. Background should mention the relevance of using IL6 as the primary end point of the study

7. It is not clear why mentioning data regarding BMD

8. Limitation should mention not using hsCRP and the possible low dose of statine which has been used (it was mentioned the fear of statines’ toxicities)

9. Line 256 which discuss the choice of using a mix linear iodel should be anticipated in statistical methods

10. Figured have low resolution and are difficult to be read.

All together the paper is well written and contributes to the research in the field of inflammaging in HIV

Reviewer #2: The authors evaluated the impact of switching from a Protease Inhibitor based regimen to raltegravir and addition of atorvastatin on metabolic disorders, inflammation and immune activation in people living with HIV.

There was no power calculation done as this was a pilot study. That in itself is not a problem as long as the authors are aware that they can only describe their results without attempting formal comparisons between groups. However the authors have tried to demonstrate differences between the two arms despite the study not being powered to do so.

Table 1: Change ‘drug users’ to either drug use or injecting drug use

Line 199: sentence relating to CD4 count should be revised for clarity

Line 201 Authors stated: Eight participants early discontinued the study protocol

This should be revised to ‘eight participants discontinued the study protocol early

Line 203: bad adherence should be changed to poor adherence

Line 224 & Table 2, line 527 authors should avoid comments like ‘no SIGNIFICANT differences were observed between groups or time points’. The study was not powered to detect differences between groups. The presentation of the results should be purely descriptive without need of any formal statistical tests to assess differences between groups

Lines 231/232: No significant changes in absolute counts or percentages of CD4 and CD8 T cells, either intragroup or intergroup were observed at any time point. The authors should remove the word significant.

I would not read too much meaning into the significant difference reported in lines 239, 254 and 259.

Line 266: The sample size is too small to examine for any meaningful interaction. OK, the authors pointed this out, so no need to even report this in the paper.

Lines 275 to 277: This should be reported as differences in lipid parameters rather than whether significant or not.

Sample is too small to detect any significant change in BMD.

The discussion of the findings should be presented bearing in mind that lack of a difference does not mean a difference does not truly exist but the study was not powered to detect any difference. Any significant differences observed with some measured parameters could have just been due to chance. I suggest the discussion is moderated accordingly.

6. PLOS authors have the option to publish the peer review history of their article (what does this mean?). If published, this will include your full peer review and any attached files.

Reviewer #1: Yes: Guaraldi Giovanni

Reviewer #2: No

---

## [Author Response · Author response to Decision Letter 0]

27 Jul 2020

Reviewer #1: 

Negredo et al present a pilot trial to assess the concomitant use of atorvastatin and raltegravir on immunological markers in treated subjects living with HIV switching from PI.

The idea to combine two potentially effective strategies options is original and merit consideration for publication, regardless negative results in reducing inflammation and immune activation.

We thank the reviewer for his/her comments.

Some issues should be considered to improve understanding of methods and overall discussion.

1. Title should mention that PLWH were PI-experienced. 

Authors have included this information in the title. 

2. Inclusion criteria should specify if all the patients were Raltegravir and statine naïve. 

The following exclusion criterium was already included in Methods: “.... and therapy with statins within the last 6 months”. Following the reviewer’s comment, authors have added raltegravir as a exclusion criteria: “.... and therapy with statins and raltegravir within the last 6 months” (Methods section). 

3. Table 1 should mention the cardiovascular risk profile of the patients and the proportion of the individuals offered statine regardless of dyslipidemia or low CVR.

Table 1 already includes information about lipid profile. In addition, the proportion of individuals with baseline LDL-cholesterol >160mg/dL, according to the study group, has been included in Table 1, following the reviewer suggestion. Only 2 (9%) patients from the Control group and 1 (5%) patient from the Raltegravir group presented LDL>160 at baseline, that was always <170 during the follow up. Unfortunately, no data about proportion of patients with hypertension are available.

4. It should be mention why a LDL cut off of 160 mg was chosen for stratification and be consistent through the paper when using mg of mmol units.

LDL cut off of 160 mg was chosen for stratification since LDL cholesterol 130-159 mg/dL is considered borderline high and >160 mg/dL is considered high for people without cardiovascular risk factors. This information has been added to the Methods section. 

All units have been changed to mg/dL in Table 1.

5. Line 284-294 is in fact part of the background and can be summarized in this section. 

This entire paragraph has been moved at the end of the Introduction to justify the purpose of the current study.

6. Background should mention the relevance of using IL6 as the primary end point of the study

A brief sentence highlighting the key role of IL-6 has been included in the Introduction section with a reference to justify it.

7. It is not clear why mentioning data regarding BMD. 

Following the suggestion of both reviewers, sentences describing changes in BMD have been removed from methods and result sections since no additional data were provided.

8. Limitation should mention not using hsCRP and the possible low dose of statine which has been used (it was mentioned the fear of statines’ toxicities). 

We have reinforced this limitation by including the reason why we used low doses of atorvastatin (to avoid toxicities) in the paragraph describing the limitations of the study.

9. Line 256 which discuss the choice of using a mix linear iodel should be anticipated in statistical methods

A brief description of methods has been included in the Methods section, as follow: “Longitudinal comparisons were performed by linear mixed effects models.”

10. Figured have low resolution and are difficult to be read. 

The resolution of figures has been improved.

All together the paper is well written and contributes to the research in the field of inflammaging in HIV

Reviewer #2: 

The authors evaluated the impact of switching from a Protease Inhibitor based regimen to raltegravir and addition of atorvastatin on metabolic disorders, inflammation and immune activation in people living with HIV.

There was no power calculation done as this was a pilot study. That in itself is not a problem as long as the authors are aware that they can only describe their results without attempting formal comparisons between groups. However the authors have tried to demonstrate differences between the two arms despite the study not being powered to do so.

Table 1: Change ‘drug users’ to either drug use or injecting drug use. 

This term has been changed. Thanks.

Line 199: sentence relating to CD4 count should be revised for clarity. 

In text, the overall mean CD4 T cells count and nadir values of all participants has been described while in Table 1, we described the mean values in each study group. Sentence has been modified for clarity.

Line 201 Authors stated: Eight participants early discontinued the study protocol. This should be revised to ‘eight participants discontinued the study protocol early. 

Authors have changed the sentences following the reviewer’s comment.

Line 203: bad adherence should be changed to poor adherence. 

Authors have changed the sentences following the reviewer’s comment.

Line 224 & Table 2, line 527 authors should avoid comments like ‘no SIGNIFICANT differences were observed between groups or time points’. The study was not powered to detect differences between groups. The presentation of the results should be purely descriptive without need of any formal statistical tests to assess differences between groups Lines 231/232: No significant changes in absolute counts or percentages of CD4 and CD8 T cells, either intragroup or intergroup were observed at any time point. The authors should remove the word significant. I would not read too much meaning into the significant difference reported in lines 239, 254 and 259.

Following the reviewer comment, the word significant has been removed when describing difference between groups throughout the manuscript.

Line 266: The sample size is too small to examine for any meaningful interaction. OK, the authors pointed this out, so no need to even report this in the paper.

The sentence describing the analysis of interaction has been removed.

Lines 275 to 277: This should be reported as differences in lipid parameters rather than whether significant or not.

Following the reviewer comment, the word significant has been removed when describing difference between groups throughout the manuscript.

Sample is too small to detect any significant change in BMD.

Following the suggestion of both reviewers, the sentences describing changes in BMD have been removed from methods (Objectives) and result sections.

The discussion of the findings should be presented bearing in mind that lack of a difference does not mean a difference does not truly exist but the study was not powered to detect any difference. Any significant differences observed with some measured parameters could have just been due to chance. I suggest the discussion is moderated accordingly.

 Discussion has been modified by removing any reference to significant changes between groups, and highlighting the minimal longitudinal changes observed (as mention in the conclusions).

---

## [Decision Letter · Decision Letter 1]

20 Aug 2020

A Randomized Pilot Trial to Evaluate the Benefit of the Concomitant Use of Atorvastatin and Raltegravir on Immunological Markers in Protease-Inhibitor-Treated Subjects living with HIV

PONE-D-20-01200R1

Dear Dr. Negredo,

We’re pleased to inform you that your manuscript has been judged scientifically suitable for publication and will be formally accepted for publication once it meets all outstanding technical requirements.

Kind regards,

Alan Winston

Academic Editor

PLOS ONE
---

## [Editor Report · Acceptance letter]

4 Sep 2020

PONE-D-20-01200R1 

A Randomized Pilot Trial to Evaluate the Benefit of the Concomitant Use of Atorvastatin and Raltegravir on Immunological Markers in Protease-Inhibitor-Treated Subjects living with HIV. 

Dear Dr. Negredo:

I'm pleased to inform you that your manuscript has been deemed suitable for publication in PLOS ONE. Congratulations! Your manuscript is now with our production department. 

Kind regards, 

on behalf of

Prof. Alan Winston 

Academic Editor

PLOS ONE